# CCL18 in the Progression of Cancer

**DOI:** 10.3390/ijms21217955

**Published:** 2020-10-26

**Authors:** Jan Korbecki, Mateusz Olbromski, Piotr Dzięgiel

**Affiliations:** 1Department of Histology and Embryology, Department of Human Morphology and Embryology, Wroclaw Medical University, Chałubińskiego 6a St, 50-368 Wrocław, Poland; olbromski.m@gmail.com (M.O.); piotr.dziegiel@umed.wroc.pl (P.D.); 2Department of Physiotherapy, Wroclaw University School of Physical Education, Ignacego Jana Paderewskiego 35 Av., 51-612 Wroclaw, Poland

**Keywords:** CCL18, chemokine, PITPNM3, tumor-associated macrophages, cancer, tumor, metastasis, regulatory T cells

## Abstract

A neoplastic tumor consists of cancer cells that interact with each other and non-cancerous cells that support the development of the cancer. One such cell are tumor-associated macrophages (TAMs). These cells secrete many chemokines into the tumor microenvironment, including especially a large amount of CCL18. This chemokine is a marker of the M2 macrophage subset; this is the reason why an increase in the production of CCL18 is associated with the immunosuppressive nature of the tumor microenvironment and an important element of cancer immune evasion. Consequently, elevated levels of CCL18 in the serum and the tumor are connected with a worse prognosis for the patient. This paper shows the importance of CCL18 in neoplastic processes. It includes a description of the signal transduction from PITPNM3 in CCL18-dependent migration, invasion, and epithelial-to-mesenchymal transition (EMT) cancer cells. The importance of CCL18 in angiogenesis has also been described. The paper also describes the effect of CCL18 on the recruitment to the cancer niche and the functioning of cells such as TAMs, regulatory T cells (T_reg_), cancer-associated fibroblasts (CAFs) and tumor-associated dendritic cells (TADCs). The last part of the paper describes the possibility of using CCL18 as a therapeutic target during anti-cancer therapy.

## 1. Introduction

It is estimated that in 2018 alone there were 18.1 million new cases of cancer and 9.5 million deaths worldwide [1]. This is the second group of diseases in the world in terms of the number of deaths, only after cardiovascular diseases [2]. Such a high percentage of cancer mortality is related to a diagnosis in a late stage of the disease. Equally important reasons are an insufficient knowledge about cancer progression and a long period of clinical testing of new drugs [3]. Twenty years ago, cancer cells were considered to be some kind of isolated element in the neoplastic tumor [4]. The vast majority of research during this period focused on the cancer processes inside the cancer cell. This was partly related to the level of advancement of the research methods and tools. However, this model changed over time [5], and the relationship between non-neoplastic and neoplastic cells in the tumor started to be studied [6,7,8,9]. In particular, the focus turned towards intercellular communication. One of the elements of this communication are chemokines [10]. Chemokines are a group of about 50 chemotactic cytokines. They are responsible for the recruitment of different cells to the tumor niche and for some mechanisms that cause the migration of cancer cells. However, only a few chemokines play an important role in cancer processes. One of them is C-C motif chemokine ligand (CCL)18. This chemokine is mainly produced in the tumor by tumor-associated macrophages (TAMs), and its level is higher in the tumor than in healthy tissue. For instance, in glioblastoma multiforme, the concentration of CCL18 is over 100 times higher than in healthy brain tissue [11,12].

CCL18 plays an important role in cancer progression. However, there are no papers available containing current information about this chemokine and its role in neoplastic processes. Hence, the aim of the present paper is to gather all the key information about the role of CCL18 in cancer.

## 2. The *CCL18* Gene and the CCL18 Protein

CCL18 is a chemokine from the β-chemokine sub-family because it has a -Cys-Cys- motif at the N-terminus. The gene for this protein is located on chromosome 17q11.2, and it has three exons spread over 7.1kb [13,14]. It encodes a 750-nucleotide-long transcript. The open reading frame for mRNA is 267 nucleotides in length [14,15]. It encodes a polypeptide of 89 amino acid residues in length. This polypeptide contains a signal peptide that which is cleaved. For this reason, the mature CCL18 protein has a weight of 78kDa and a length of 69 amino acids [15,16,17] (Figure 1). The actively biological form of CCL18 can be truncated to a 68-amino-acid form without a terminal alanine at the C-terminus. CCL18 has a sequence homology of 59% with the protein and of about 50% with the cDNA for CCL3/macrophage inflammatory protein 1α (MIP-1α) [13,14,15,18]. Therefore, it is postulated that the *CCL18* gene arose from duplication and the subsequent fusion of two *MIP-1α*-like genes, hence its occurrence in primates but not in rodents [18]. In the late 1990s and early 2000s, CCL18 was characterized and named on the basis of organ-specific expression and synthesis by specific cell types [14,15,18]. As a result, four alternative names for it existed simultaneously:
-macrophage inflammatory protein 4 (MIP-4)-pulmonary and activation-regulated chemokine (PARC)-alternative macrophage activation-associated C-C chemokine-1 (AMAC-1)-dendritic cell-derived C-C chemokine 1 (DCCK1)

In particular, CCL18 is expressed in the lung, from which it owns one of its names [13,14]. In allergic asthmatics, CCL18 is produced in the lungs under the influence of allergens. It causes attracted basophil and induced basophil histamine release [19]. This process is important in the pathogenesis of asthma.

CCL18 is also produced in the germinal centers of the secondary lymphoid organs by the dendritic cells (DCs) [13,20]. It is a chemoattractant for CD38^-^IgG^-^ mantle zone B lymphocytes and CD45RA^+^ naïve T lymphocytes, but not for CD45RO^+^ memory T cells, monocytes, granulocytes, and mature DCs [13,14,18,20,21], although it is a chemoattractant for 3- to 4-day-old monocytes [22]. Due to CCL18′s recruitment of lymphocytes in secondary lymphoid organs, this chemokine is crucial for the initiation of the immune response [23].

CCL18 is also a macrophage M2c subset marker [24,25]. Interleukin (IL)-4 and IL-10 cause an increase in the production of CCL18 in macrophages [15,24,25] However, thanks to its immunosuppressive properties, CCL18 is not only a marker but also the responsible for the properties of M2c macrophages [26]. CCL18 also plays a regulatory role in inflammatory reactions, protecting against acute pro-inflammatory responses. For this reason, phorbol 12-myristate 13-acetate (PMA) and lipopolysaccharide (LPS) also increase the expression of CCL18 in monocytes [13,22].

So far, a few receptors for CCL18 have been identified. The most important of them in neoplastic diseases is the phosphatidylinositol transfer protein 3 (PITPNM3)/PYK2 N-terminal domain-interacting receptor 1 (Nir1). This receptor is involved in migration and cell metastasis in many types of cancer [27,28,29,30]. Another receptor for the CCL18 chemokine is the C-C motif chemokine receptor (CCR)8. The activation of this receptor by CCL18 also causes the migration of tumor cells [31,32]. CCL18 can also activate CCR6 on lung fibroblasts, which is important in the development of pulmonary fibrosis [33]. The next receptor for CCL18 is the G protein-coupled estrogen receptor 1 (GPER1)/G-protein coupled receptor 30 (GPR30) [34], which is also a receptor for 17β-estradiol. The activation of this receptor by CCL18 does not result in signal transduction. Instead, it only disturbs the signal transmission from the receptor for the C-X-C motif chemokine ligand (CXCL)12/stromal-derived factor-1 (SDF-1). This reduces the chemotaxis and proliferation of tumor cells with a high expression of the CXCL12/SDF-1 chemokine [35]. It has been proven that CCL18 has this type of influence on pre-B acute lymphocytic leukemia cells [34]. Another example of receptor to which CCL18 binds is CCR3 [36,37]. Nevertheless, CCL18 exhibits an antagonistic activity in regard to this receptor, which translates into the inhibition of the infiltration of activated eosinophils, Th2 lymphocytes and basophils. CCL18 activates or has an influence on the function of many different receptors. However, only a few papers are available on the importance of receptors other than PITPNM3 in the action of CCL18. In order to better understand the role of CCL18 in neoplastic processes, additional research on the significance of the activation of the abovementoned receptors by CCL18 is required.

## 3. CCL18 in Tumor Progression

In vitro [6,7,38,39] and in vivo studies [16,40,41,42,43,44] have shown that, in the tumor, CCL18 is produced in large amounts by TAMs, and also in smaller amounts by cancer-associated fibroblasts (CAF) [45] and cancer cells such as colon cancer cells [46], glioblastoma multiforme cells [40], non-small cell lung cancer cells [7], melanoma cells [47], and renal cell carcinoma cells [48]. The expression of CCL18 in neoplastic cells is increased by the WNT→β-catenin pathway, as shown by in vitro experiments in the colon cancer [46] and renal cell carcinoma models [48]. This means that the effect of CCL18 in the tumor appears after TAM recruitment to the tumor niche.

The mechanism of the pro-tumor activity of CCL18 can be divided into two aspects. First, it has a direct effect on the cancer cell, in particular through the PITPNM3 receptor. As a result, the migration of neoplastic cells is induced [27]. The second aspect is the effect on non-cancerous cells, which cooperate with the neoplastic cells in the tumor. The greatest amounts of CCL18 are produced by TAMs [7,42], with CAFs producing smaller amounts [45]. This chemokine acts on these cells in an autocrine manner. For this reason, CCL18 is essential in the interdependence of TAMs and CAFs, as well as in the influence of these cells on regulatory T cells (T_reg_) [49,50] and tumor-associated dendritic cells (TADCs) [26].

### 3.1. Effect of CCL18 on Cancer Cells

#### 3.1.1. Influence of CCL18 on Cancer Cell Proliferation

CCL18 may have an influence on the proliferation of cancer cells. However, this effect is dependent on the type of tumor. In non-small cell lung cancer cells, CCL18 reduces proliferation [51]. It also reduces the proliferation of pre-B acute lymphocytic leukemia cells [34]. This effect depends on the activation of GPER1/GPR30, which leads to the disruption of the function of CXCR4. In breast cancer [52,53], diffuse large B cell lymphoma [54], ovarian cancer [55], osteosarcoma [56], and urothelial carcinoma [57], the chemokine increases the proliferation of cancer cells. Moreover, CCL18 does not affect the proliferation of gastric cancer [58] and pancreatic ductal adenocarcinoma cells [59]. At least in osteosarcoma cells, CCL18 causes an increase in the expression of the urothelial carcinoma associated 1 (UCA1), which results in the activation of the WNT→β-catenin pathway [56]. This in turn results in an increased proliferation and migration of these cells. It has also been shown that CCL18 can increase the expression of stem cell markers in the cells of tumors such as oral squamous cell carcinoma [60] and squamous cell carcinoma of the head and neck [61]. In the oral squamous cell carcinoma HSC-6 and CAL33 cell lines, this was associated with the activation of the mammalian target of rapamycin (mTOR) [60].

#### 3.1.2. CCL18 as an Inducer of EMT and Migration of Tumor Cells

One of CCL18′s receptors is PITPNM3. This receptor has been best studied in CCL18-dependent migration induction, invasion and epithelial-to-mesenchymal transition (EMT) tumor cells. PITPNM3 is important in the migration and metastasis of cells of tumors such as hepatocellular carcinoma [62], breast cancer [27,63,64,65], non-small cell lung cancer [30,51], oral squamous cell carcinoma [29], ovarian cancer [66], pancreatic ductal adenocarcinoma [67], prostate cancer [28], and squamous cell carcinoma of the head and neck [68].

The activation of PITPNM3 causes signal transduction through several pathways (Figure 2). Phosphorylation of the phospholipase Cγ1 (PLCγ1) and the protein kinase C-ζ (PKCζ) [69] takes place, as well as an increase of the expression of the inositol 1,4,5-trisphosphate 3-kinase isoform B (IP3KB), which activates intracellular calcium signaling [69]. PITPNM3 causes signal transduction through the JAK2→STAT3 pathway, leading to the proliferation, migration, and EMT of oral squamous cell carcinoma cells [70].

PITPNM3 causes signal transduction through the proline-rich tyrosine kinase 2 (Pyk2) [27,62,66]. Pyk2 is a non-receptor protein tyrosine kinase belonging to the focal adhesion kinase (FAK) family [71]. An activated Pyk2 causes the phosphorylation of a multiple-domain Arf-GAP protein 1 (AMAP1) (other names: DDEF1 and ASAP1) [64]. This causes the dissociation of the AMAP1 complex from the inhibitor of NF-κB kinase β subunit (IKKβ). IKKβ is released, and then the nuclear factor κB (NF-κB) is activated. NF-κB is directly responsible for the transcription of the genes responsible for EMT [72,73,74]. As a result, the migration and metastasis of neoplastic cells take place [62]. The activation of NF-κB by CCL18 also increases the expression of metadherin (MTDH), which leads to the EMT of the cells of the squamous cell carcinoma of the head and neck [68]. However, the CCL18-dependent activation of NF-κB can be inhibited by IL-32θ in breast cancer cells [75]. This is related to the interaction of IL-32θ with protein kinase C-δ (PKCδ), although the importance of PKCδ in the effects of CCL18 on the tumor cell still needs to be thoroughly investigated.

Activated by PITPNM3, Pyk2 can activate Src in breast cancer cells [76]. This kinase is essential in integrin α_5_β_1_ clustering-dependent adherence. As a result, the integrin activation and migration of the cancer cells take place. The signal transmission also includes engulfment and cell motility 1 (ELMO1), which has been proven on non-small cell lung cancer cells [30]. This protein causes β_1_ integrin phosphorylation and the activation of RAC1, which is also involved in the polymerization of actin and the migration of cancer cells.

When PITPNM3 is activated, the phosphatidylinositol-4,5-bisphosphate 3-kinase (PI3K)→Akt/protein kinase B (PKB) pathway is also activated [29,63,75]. In breast cancer cells, this is related to the activation of the Src kinase by Pyk2 [76]. Annexin A2 plays an important role in this mechanism [77]. However, it is also possible that the activation of NF-κB is dependent on the PI3K→Akt/PKB pathway. This pathway causes the activation of NF-κB, which produces an increase in the expression of Lin28b in breast cancer cells [78]. This leads to a reduction in the levels of miR-98 and miR-27b. On the one hand, miR-98 silences N-Ras. A reduction in the levels of miR-98 lead to the activation of the N-Ras→extracellular signal-regulated kinase (ERK) mitogen-activated protein kinase (MAPK)→PI3K→NF-κB→Lin28b pathway. On the other hand, miR-27b inhibits EMT in cancer cells. When miR-27b levels are reduced, EMT is induced in breast cancer cells.

The activation of Akt/PKB leads to the activation of two pathways [63]. The first one, LIN-11, Isl1 and MEC-3 protein domain kinase (LIMK)→cofilin, is responsible for the polymerization of actin and, therefore, the migration of cancer cells. In the second pathway, Akt/PKB causes the phosphorylation of glycogen synthase kinase-3β (GSK-3β). This reduces the activity of GSK-3β, which leads to the stabilization of Snail. Snail levels increase, leading to EMT [59,63,77,79].

In breast cancer cells, CCL18 also activates the p300/CBP-associated factor (PCAF). This is the acetyltransferase that causes the acetylation of ACAP4 [80] and ezrin [81]. Acetylated ACAP4 regulates membrane cytoskeletal dynamics, which results in cell migration. In contrast, ezrin acetylation is associated with directionally persistent migration. Nevertheless, there are no data on the exact way of activation of the PCAF by CCL18.

Additionaly, CCL18 is proved to play a crucial role in the migration and EMT processes of tumor cells by activating receptors other than PITPNM3. In particular, CCL18 promotes the EMT activation of blader cancer cells by activating CCR8 [32]. However, CCL18 may also reduce cancer cell migration by activating GPER1/GPR30, where this chemokine interferes with the action of CXCR4 on pre-B acute lymphotic leukemia cells [34]. This reduces the migration of these cells in response to CXCL12/SDF-1.

CCL18 can activate other signaling pathways, but some of them have not yet been associated with any of the receptors. In some types of cancer, CCL18 can activate mTOR. The activity of mTOR in neoplasms is often deregulated, which leads to increased proliferation, migration of cancer cells and an increase in the synthesis of nucleotides and proteins [82]. mTOR activation via CCL18 leads to cell migration in tumors such as endometrial cancer [83], ovarian cancer [55], and oral squamous cell carcinoma [60]. CCL18 seems to activate mTOR via the PI3K→Akt/PKB pathway, as shown by experiments on endometrial cancer cells [83]. The activation of mTOR leads to an increase in KIF5B expression in the cell line Ishikawa (endometrial cancer), which, in turn, leads to EMT [83]. The activation of mTOR also increases Slug expression and the induction of EMT in oral squamous cell carcinoma, as demonstrated by experiments conducted on the HSC-6 and CAL33 cell lines [60]. However, an increase in Slug expression can occur by the activation by CCL18 of the ERK MAPK→NF-κB cascade, which was demonstrated on MGC-803 gastric cancer cells [58].

CCL18 also causes the migration of the osteosarcoma cell lines MG63 and 143B [56]. This action is dependent on the increase in UCA1 expression by the EP300 transcription factor. As a consequence, the WNT→β-catenin pathway is activated, which results in the proliferation and migration of osteosarcoma cells.

Nonetheless, many of the aforementioned data have been obtained without the association of these pathways to a given receptor for CCL18. Therefore, in order to better understand the effect of this chemokine, research into the exact mechanism of action of CCL18 and the study of which receptor is activated by it are required. In addition to PITPNM3, other receptors are also responsible for the CCL18-dependent migration and EMT of tumor cells, for instance CCR8 in bladder cancer cells [32].

#### 3.1.3. Influence of microRNA in the Function of CCL18

CCL18 expression and function is under the control of microRNAs. Particularly, CCL18 is a target gene of miR-128 [84], miR-205 [47], and miR-622 [85]. In melanoma cells, colorectal neoplasia differentially expressed (CRNDE) long non-coding RNA (lncRNA) regulates miR-205, which causes CCL18 expression to increase [47].

MicroRNAs also regulate the function of CCL18. miR-622 blocks the activation of ERK MAPK, thereby disrupting the activation of this cascade by CCL18, and thus the action of this chemokine [85]. Moreover, miR-181b decreases NF-κB expression, which interferes with the function of CCL18, as this transcription factor is important in the induction of migration and EMT by CCL18 [52].

CCL18 also causes changes in microRNA expression, which causes the migration of cancel cells. The activation of NF-κB by CCL18 results in an increase in the expression of Lin28b in breast cancer cells [78]. This leads to a reduction in the levels of miR-98 and miR-27b. MiR-98 silences N-Ras, i.e., CCL18 causes the activation of the N-Ras→ERK MAPK→PI3K→NF-κB→Lin28b pathway, which increases the activity of CCL18. On the other hand, miR-27b inhibits EMT in cancer cells. When miR-27b levels are reduced, EMT of breast cancer cells is induced.

In esophageal squamous cell carcinoma, CCL18 increases the expression of the hox transcript antisense intergenic RNA (HOTAIR) [86]. This is a lncRNA that functions as a miR-130a-5p sponge. This results in a decrease in the level of miR-130a-5p, and therefore an increase in the level of ZEB1. As a consequence, EMT of cancer cells takes place.

### 3.2. Influence of CCL18 on Tumor-Associated Cells and Tumor Microenvironment

#### 3.2.1. Effect of CCL18 on Angiogenesis and Lymphangiogenesis

In the initial stages, the intensive proliferation of neoplastic cells and tumor growth are not related to the development of blood vessels [87]. This causes hypoxia in the centre of the growing tumor, which results in significant changes in its functioning. Angiogenesis is closely related to the migration and metastasis of neoplastic cells [88,89] takes place. One of the most important factors inducing angiogenesis is the vascular endothelial growth factor (VEGF), although there are also numerous other factors that induce angiogenesis in the tumor. One of them is CCL18 [50,90]. This is associated with the presence of the PITPNM3 receptor on the cells of the blood vessel walls. Studies with human umbilical vein endothelial cells (HUVECs) show that CCL18 causes the VEGF-independent migration and tube formation of these cells [90]. CCL18 also causes HUVECs EMT, which is dependent on ERK MAPK activation and the Akt/PKB→GSK-3β→Snail pathway, as well as EMT of cancer cells [59,63,77]. At the same time, in the bladder cancer model, CCL18 causes an increase in the production of VEGF-C and matrix metalloproteinase-2 (MMP-2), which are factors involved in lymphangiogenesis [32]. This process is dependent on CCR8.

*CCL18* is a hypoxia-repressed gene [91,92,93]. Under chronic hypoxia conditions, the expression of CCL18 is independent of the hypoxia inducible factors (HIFs) activation but dependent on the lysine-specific demethylase 6B (KDM6B)/Jumonji domain-containing protein D3 (JMJD3) activity [94]. This enzyme is a histone demethylase, which is an oxygen-dependent enzyme. A reduction in oxygen concentration causes a decrease in the activity of KDM6B/JMJD3, which results in histone methylation and thus a decrease in CCL18 expression at the transcription level.

#### 3.2.2. Tumor-Associated Macrophages and CCL18 in the Neoplastic Tumor

A neoplastic tumor not only contains cancer cells. There are also tumor-associated cells that participate in the progression of the tumor. One of these cells are TAMs [95]. An increased number of these cells in the neoplastic tumor correlates with a worse prognosis for patients of multiple neoplasms [96,97,98]. These cells are derived from monocytes, which are recruited into the tumor niche and then differentiated into TAMs. Certain chemokines, such as the CCL2/monocyte chemoattractant protein (MCP)-1 [99], the CCL5/regulated on activation, the normally T cell expressed and secreted (RANTES) [100] and the CCL8/MCP-2 [101] (Figure 3) are responsible for the recruitment of TAMs. However, CCL18 is not a chemotactic agent for monocytes or macrophages [13,14,18,20,21], which is why it does not affect the recruitment of TAMs into the tumor niche. Nevertheless, TAMs are responsible for the production of CCL18 in the tumor [6,7,16,38,39,40,41,42,43,44].

TAMs in human tumors are not strictly prescribed to the M2 subset [9,42]. They show mixed M1 and M2 phenotype. Examples of factors causing the polarization of macrophages in the tumor microenvironment are IL-4 and IL-6 [102], prostaglandin E_2_ (PGE_2_) [103], IL-10 released by T_reg_ cells and the direct cell–cell interaction of these cells with macrophages [15,104], tumor acidification [105,106], extracellular matrix [107], granulocyte-macrophage colony-stimulating factor (GM-CSF) [108], connective tissue growth factor (CTGF) [109], and many others. CCL18 affects the phenotype of TAMs. In the absence of other differentiating factors, CCL18 causes the differentiation of monocytes into M2 macrophages [110]. These macrophages show the expression of the M2 polarization marker CD206, as well as the expression of cytokines that are important in the progression of cancer: IL-10, CXCL8/IL-8 and CCL2/MCP-1. CCL18 also increases the production of the CXCL1/growth related oncogene-α (GRO-α) and IL-6 in macrophages [111]. Some of these chemokines are involved in the recruitment of cells into the tumor niche: CCL2/MCP-1 in the recruitment of TAMs [99], and CXCL1/GRO-α in the recruitment of neutrophils [112] and myeloid-derived suppressor cells (MDSC) [113]. These chemokines also participate in the angiogenesis and migration of cancer cells [114,115].

In vitro studies have shown that CCL18 is mainly produced by TAMs in the neoplastic tumor [6,7,16,38,39,40,41,42,43,44]. CCL18 expression in macrophages is increased by estrogen receptor α (ERα) activation [83], IL-4 and IL-6 [102], extracellular matrix [107], IL-10 released by T_reg_ cells and the direct cell–cell interaction of these cells with TAMs [104], GM-CSF [108], and CTGF [109]. In contrast, CCL18 expression in macrophages is reduced by interferon-γ (IFN-γ) [116,117]. In a tumor, CCL18 also indirectly influences TAMs and thus its own synthesis. NF-κB activated by CCL18 is equally responsible for the increase in the expression of the vascular cell adhesion molecule-1 (VCAM-1) on pancreatic ductal adenocarcinoma cells [67]. This protein causes TAMs to bind to the cancer cells. TAMs produce CCL18, increasing the production of VCAM-1 even further. Additionally, this protein increases the Warburg effect in the cancer cell. This in turn increases the secretion of lactate into the neoplastic microenvironment, which causes its acidification. This acidification induces alternative activated M2 phenotype macrophages polarization and, thus, the production of CCL18 in TAMs is further increased [67,105,106]. Many chemokines are also under the control of NF-κB. For this reason, the activation of this transcription factor by CCL18 causes an increase in the expression of the chemokines responsible for the recruitment of more TAMs. This leads to an increase in the production of CCL18 in the vicinity of the tumor cells [108].

#### 3.2.3. CCL18 and T_reg_ Recruitment into the Tumor Niche

T_reg_ cells are responsible for cancer immune evasion because they inhibit the anti-tumor immune response [118]. As shown in breast cancer research studies, these cells in the tumor niche express CCR8 (receptor for CCL1/I-309) and CCR4 (receptor for CCL17/thymus and activation regulated chemokine (TARC) and CCL22/macrophage derived chemokine (MDC)) [119,120]. For this reason, chemokines such as CCL1/I-309 (in breast cancer) [121,122], CCL17/TARC, CCL22/MDC (in gastric cancer and hepatocellular carcinoma) [123,124], and CCL28/MEC (in hepatocellular carcinoma) [125] cause T_reg_ recruitment into the tumor niche. T_reg_ cells are also recruited by the CCL20/macrophage inflammatory protein 3α (MIP-3α)→CCR6 axis, as shown in research on colorectal cancer [126], hepatocellular carcinoma [127], and non-small-cell lung carcinoma [128] models.

CCL18, produced by TAMs, also plays an important role in T_reg_ recruitment [50,116,129]. The CCL18 chemokine recruits CD4^+^CD45RA^+^CD25^-^ naïve T cells into the tumor niche, and then they are differentiated into T_reg_ cells, as shown in research studies on gastric cancer [130] and breast cancer [129]. This process is related to the recruitment of these cells via the PITPNM3 receptor [129]. Then, the differentiation of naïve T cells into T_reg_ cells takes place. Next, in the neoplastic tumor, there is a mutual interaction between TAMs and T_reg_ cells. On the one hand, TAMs participate in the recruitment of T_reg_ cells through CCL18 [129,130], while on the other hand, T_reg_ cells increase macrophage polarization and the production of CCL18 in them [104]. This is related to the production of IL-10 by T_reg_ and the direct cell–cell interaction of these cells with TAMs.

CCL18 also participates in the recruitment of memory CD4^+^CD25^+^Foxp3^-^ T cells [49,131]. These cells produce IL-4, IL-10, and the transforming growth factor β1 (TGF-β1), which are factors involved in neoplastic processes. CCL18 can also convert CD4^+^CD25^-^ memory T cells into CD4^+^CD25^+^Foxp3^+^ T_reg_ [132]. The recruitment of naïve T cells by CCL18 has been confirmed in tumor models [129,130]. However, the recruitment of memory T cells by this chemokine has not yet been proven in these models [49,132].

The mechanisms of T_reg_ recruitment into the tumor niche are not yet fully understood [131,133]. T_reg_ cells can be divided into over 20 subsets [134]. It is likely that many types of T cells are recruited into the tumor niche. Many chemokines involved in this process, not only CCL18, are also produced by TAMs in the neoplastic tumor [25,42,108]. Examples of such chemokines are CCL17/TARC and CCL22/MDC [123,124]. CCL18 may be involved in recruiting certain types of T cells, and it can also participate in differentiating them into T_reg_ cells. However, more research is required in order to better understand T_reg_ recruitment into the tumor niche.

#### 3.2.4. Tumor-Associated Dendritic Cells and CCL18

DCs are cells involved in the normal anti-tumor immune response. Mature DCs are recruited into the tumor niche by the CCL19/MIP-3β and CCL21/secondary lymphoid tissue (SLC) chemokines [135]. They stimulate cytotoxic NK cells [136,137] and CD8^+^ T lymphocytes [138] to kill cancer cells. DCs also kill cancer cells directly [139]. On the other hand, immature DCs are recruited by CCL20/MIP-3α [135,140]. CCL18 also causes the chemotaxis of immature DCs, which shows that this chemokine may be involved in the direct recruitment of immature DCs into the tumor niche [141] (Figure 4). In the tumor microenvironment, these cells are differentiated into TADCs. These cells have immunosuppressive characteristics because they participate in the recruitment of T_reg_ cells into the tumor niche and enhance their pro-cancer properties [142]. They also produce the heparin-binding EGF-like growth factor (HB-EGF) and amphiregulin (epidermal growth factor receptor (EGFR) ligands), thus stimulating the development of lung cancers [143,144]. In colon cancer, TADCs also produce CCL5/RANTES [145]. This chemokine participates in the migration of cancer cells and the recruitment of various types of cells into the tumor niche. However, the exact pro-cancer effect of TADCs is still poorly understood and requires further thorough research.

The factors influencing the formation of TADCs in the neoplastic tumor have not yet been studied in depth. However, it seems that the most important of them are PGE_2_ [146], CCL18 [26], and IL-10 [147]. In the tumor, CCL18, IL-10, and PGE_2_ are produced by TAMs. CCL18 can also be produced by immature DCs [35,141,148]. CCL18 causes the chemotaxis of immature DCs into the tumor niche [141]. At this point, CCL18 causes an increase in IL-10 expression in the differentiating DCs [26]. IL-10 then causes an increase in the expression of indoleamine 2,3-dioxygenase (IDO) [147]. The same goes for PGE_2_, which, from the tumor microenvironment, increases the expression of IDO in the differentiating DCs [146]. The expression of IDO in the DCs causes an increase in the number of T_reg_ cells [26,149,150,151]. The differentiation of naïve CD4^+^ T cells into T_reg_ cells is behind this process [142]. CCL18 is also important in this process, because it is responsible for the chemotaxis of naïve CD4^+^ T cells close to TAMs and immature DCs [20,23,130].

#### 3.2.5. Significance of the Effects of CCL18 on CAFs in the Neoplastic Tumor

CAFs are the cells in the tumor niche responsible for the production and remodeling of the extracellular matrix [152,153,154]. They also secrete many factors, such as chemokines that support tumor growth. In breast phyllodes tumor, CCL18 participates in myofibroblast differentiation [39,155]. This process is dependent on the PITPNM3 receptor and the NF-κB→miR-21→tumor suppressor phosphatase and tensin homolog (PTEN)→Akt/PKB pathway, which is activated by this receptor [39]. As a result, the differentiating myofibroblasts begin remodeling the extracellular matrix, which has been proven on a collagen contraction assay. However, there are no detailed studies on the effects of CCL18 on CAFs in the neoplastic tumor.

CCL18 is a factor involved in the development of pulmonary fibrosis [156]. This chemokine increases the production of collagen in the fibroblasts [156,157,158]. This process may be CCR6-dependent [33]. The increase in the amount of collagen is also important in neoplastic processes [152,153,159,160]. Nevertheless, there are no studies available on the influence of CCL18 on the production of collagen in the neoplastic microenvironment.

#### 3.2.6. Role of Glycosaminoglycans in the Mechanisms of Action of CCL18

Chemokines are a group of chemotactic cytokines that cause the activation of their receptors and the chemotaxis and migration of various types of cells. However, free chemokines have a poor ability to induce chemotaxis. Only after binding with glycosaminoglycans (GAGs) on the cell surface, do they show full activity [21,161,162]. Nevertheless, GAGs have a limited number of chemokine binding sites. For this reason, some chemokines that are less bound to GAGs are displaced by other chemokines that bind more strongly to them. An example of a chemokine that binds strongly to GAGs is CCL18 [37]. The high concentration of CCL18 in the neoplastic tumor causes the displacement of GAG-bound chemokines, which disrupts their functioning. CCL18 has been shown to interfere with 16 of the 22 tested chemokines from the CC sub-family, as well as with 12 of the 15 chemokines tested from the CXC sub-family [37]. However, CCL18 displaces different chemokines to varying degrees. For instance, CCL2/MCP-1 and CCL5/RANTES are not significantly replaced by CCL18 [37]. These are important chemokines in the neoplastic processes, involved in the recruitment of TAMs into the tumor niche, angiogenesis and metastasis [114]. The chemokines more strongly displaced by CCL18 are CXCL14/breast and kidney chemokine (BRAK), CCL1/I-309, CCL11/Eotaxin-1, CCL19/MIP-3β, CCL25/thymus expressed chemokine (TECK), and CCL26/Eotaxin-3. CXCL14/BRAK is an angiostatic and anti-tumor chemokine [163,164,165]. CCL11/Eotaxin-1, by recruiting eosinophils into the neoplastic niche, also has anti-tumor properties [166,167]. Therefore, CCL18 inhibits the action of some anti-cancer chemokines and, to a much lesser extent, pro-cancer ones.

CCL18 binding to GAGs has another important role in neoplastic processes. It allows extracellular vesicles to attach to target cells in the tumor microenvironment [168]. Extracellular vesicles are tiny membrane vesicles that are involved in extracellular communication [169]. They allow the transfer of miRNAs, mRNAs, and various proteins between cells. For this reason, they play an important role in neoplastic processes, in particular angiogenesis, migration, and metastasis, as well as in cancer immune evasion. They are secreted by different cells in the tumor niche, especially by tumor cells, and then they must be attached to the target cell. CCL18 plays a crucial role in this process [168]. This chemokine binds to GAGs on the extracellular vesicles. Next, GAG-bound CCL18 binds to its CCR8 receptor on the target cell. This allows extracellular vesicles to attach to the target cell, which is important in intercellular communication involving extracellular vesicles.

## 4. CCL18 as a Tumor Marker

CCL18 is a marker of neoplastic diseases. Studies on patients’ material have shown that CCL18 expression is higher in tumor tissues than in normal ones in tumors such as cutaneous basal cell carcinoma [170], glioma [11,12], and breast cancer [171]. Additionally, the serum concentration of CCL18 is higher in cancer patients than in healthy individuals, especialy in cases of T-cell acute lymphoblastic leukemia [6], breast cancer [172,173], cutaneous T-cell lymphoma [148], laryngeal squamous cell carcinoma [174], non-small cell lung cancer [175,176], ovarian cancer [177], pancreatic ductal adenocarcinoma [59], and prostate cancer [178]. Moreover, in bladder cancer patients [179], the urinary concentration of CCL18 is higher than in healthy individuals. The level of CCL18 is proportional to the level of disease development: stage, Ki67 expression level in the tumor and lymph node metastasis [12,171,172,175,180,181]. Increased levels of CCL18 in the serum or the tumor are also associated with a worse prognosis for patients of cancers such as breast cancer [39,69,108,173], cutaneous T-cell lymphoma [148], laryngeal squamous cell carcinoma [174], lung cancer [175,176], oral squamous cell carcinoma [182], ovarian cancer [55,177], osteosarcoma [56], pancreatic ductal adenocarcinoma [59,67], and esophageal squamous cell carcinoma [86]. An increased level of CCL18 in colorectal cancer [43] and gastric adenocarcinoma [44] is associated with a better prognosis. In non-small cell lung cancer, a greater infiltration of CCL18-producing cells is associated with a better prognosis [183], although the level of this chemokine in the plasma of patients with this type of cancer is associated with a worse prognosis [175] (Table 1). Nevertheless, the data in the Human Protein Atlas (https://www.proteinatlas.org) on the level of mRNA for CCL18 in tumors of various cancers do not confirm a clear relationship between an increased CCL18 expression and a worse survival [184,185] (Table 2). Taking into consideration 17 types of neoplasms, only in 4 cases an increased level of CCL18 is associated with a worse prognosis for patients, and in other 4, with a better one. However, there are no available data on the anti-cancer properties of CCL18. It is likely that this chemokine causes an increase in tumor-infiltrating lymphocytes (TILs) in the tumor [23], which, with an effective anti-tumor immune response, may improve the patient’s prognosis [186].

## 5. CCL18 as a Therapeutic Target in Anti-Cancer Therapy

CCL18 is produced in large amounts in every tumor [6,12,170,175,178], where it participates in the recruitment of T_reg_ cells [50,116,129] and affects the phenotype of TAMs [110]. CCL18 is also involved in the migration, invasion and metastasis of many neoplastic tumors, although it is its role in the metastasis of breast cancer that has been best studied [27,63,64,65]. For this reason, understanding how CCL18 acts can be useful for anti-cancer therapy. Furthermore, the expression of CCL18 in TAMs may be determinant in the potential therapeutic approach. It has been shown that, in the presence of YKL-39^+^CCL18^-^ TAM or YKL-39^-^CCL18^+^ TAM in a tumor, patients did not respond to neoadjuvant chemotherapy [187]. Nevertheless, more research is required to link CCL18 expression with the response to anti-cancer theraphy.

CCL18 is a potential therapeutic target for anti-cancer therapy. However, to date (second half of 2020), there is only one study focusing on the effects that blocking the activity of CCL18 would have on tumor development. Said study used SMC-21598 [65]. This is a compound that binds to CCL18, causing the chemokine to lose its biological properties. In vivo studies on NOD/SCID mice inoculated with MDA-MB-231 breast cancer cells showed that blocking CCL18 activity does not affect tumor growth or metastasis to the lungs [65]. This compound only blocks metastasis induced by CCL18 administration. Nevertheless, metastasis is a multi-factorial process, and more research is required in order to explore the correct therapeutic approach properly. Specifically, research is needed on the simultaneous application of multiple therapeutic approaches, such as blocking the activity of CCL18 and other pro-cancer factors. Hopefully, such studies will appear in the coming years.

Currently, no experimental work examining CCL18 as a therapeutic target in cancer other than breast cancer is available. However, studies on patients’ prognosis have shown that an increased expression of CCL18 in tumors such as esophageal squamous cell carcinoma [86], glioma [185], oral squamous cell carcinoma [182], ovarian cancer [55,177], pancreatic cancer [67,185], prostate cancer [185], and urothelial cancer [185] is associated with a worse prognosis. This indicates that CCL18 plays an important pro-tumor role in these types of cancer. For this reason, future research on these diseases should mainly focus on developing therapeutic approaches that target CCL18.

## Figures and Tables

**Figure 1 ijms-21-07955-f001:**
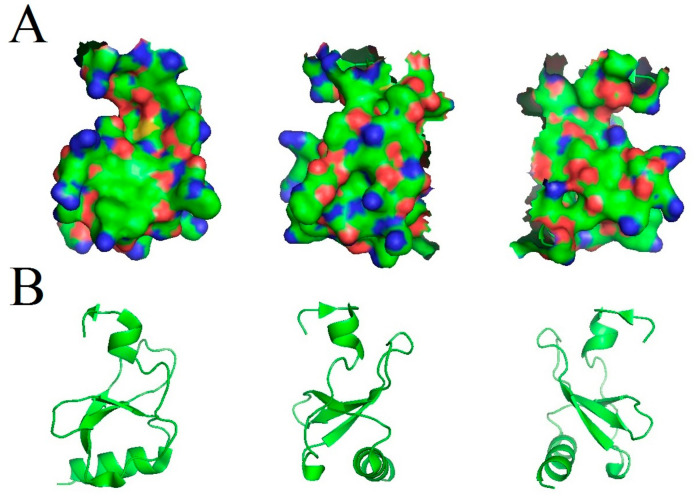
Modeled structure of the CCL18 protein shown under different angles. (**A**) Three-dimensional structure of the CCL18 protein. (**B**) Third-order structure of the CCL18 protein. Source: PDB ID: 4mhe; [17].

**Figure 2 ijms-21-07955-f002:**
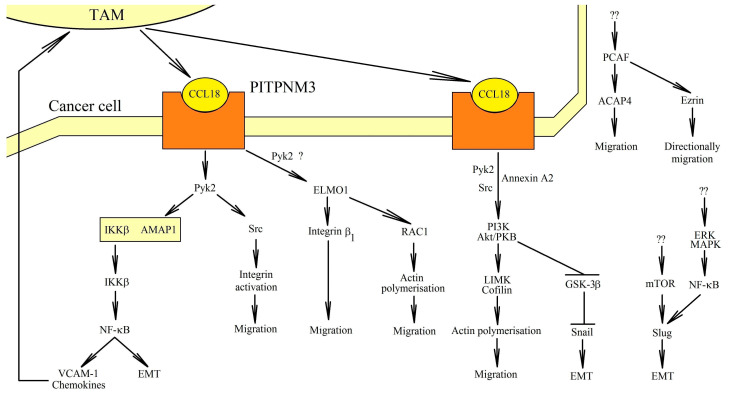
Role of PITPNM3 in cancer cell migration. CCL18 activates its PITPNM3 receptor. This results in signal transduction from this receptor. Pyk2 activation takes place, which leads to the activation of NF-κB. This transcription factor causes EMT. NF-κB also increases the expression of chemokines and VCAM-1, which cause the recruitment of TAMs in the vicinity of the tumor cell and, consequently, an even greater concentration of CCL18 in the vicinity of the PITPNM3 receptor. In addition to the NF-κB-dependent pathway, PITPNM3 causes the activation of ELMO1, which leads to changes in actin polymerization and tumor cell migration. At the same time, with the activation of Pyk2, the PI3K→Akt/PKB pathway is activated, leading to an increase in Snail expression and then to the EMT of tumor cells. CCL18 can also activate other signaling pathways, but some of them have not yet been associated with any of the receptors. CCL18 can activate mTOR, which leads to an increase in Slug expression. This leads to the EMT of tumor cells. CCL18 also causes directional migration of breast cancer cells by increasing ezrin acetylation by PCAF.

**Figure 3 ijms-21-07955-f003:**
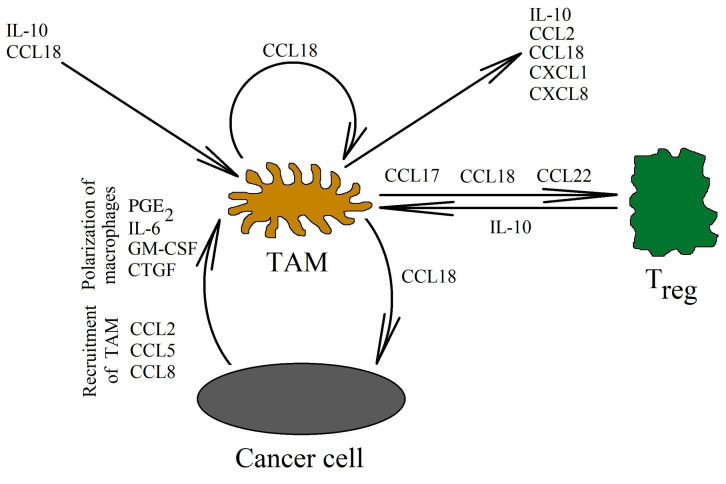
CCL18 as an important factor in the interaction of TAMs with cells in the neoplastic tumor. CCL18 is mainly produced by TAMs. These cells are recruited into the tumor niche as monocytes by different chemokines, but not by CCL18. Subsequently, monocytes undergo differentiation into TAMs by various factors from the tumor microenvironment (one of them is CCL18). Next, CCL18 expression induction takes place, which causes T_reg_ cells recruitment into the tumor niche. This chemokine also increases the expression of factors such as IL-10, CCL2/MCP-1, CXCL1/GRO-α, and CXCL8/IL-8. These factors are involved in the development of the neoplastic tumor by recruiting tumor-associated cells, and they also participate in the migration of neoplastic cells and in cancer immune evasion.

**Figure 4 ijms-21-07955-f004:**
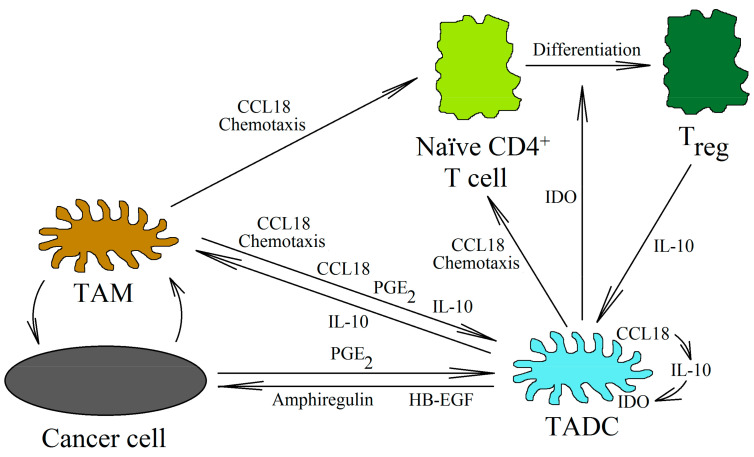
Tumor-associated dendritic cells in the tumor niche. Immature DCs and TAMs produce CCL18 in the tumor niche. This chemokine causes the chemotaxis and recruitment of immature DCs and naïve CD4^+^ T cells into the tumor niche. Next, CCL18 causes the differentiation of immature DCs into TADCs. The production of IL-10 increases in these cells, and thus the expression of IDO is autocrinally increased. However, PGE_2_, produced by various cells (including TAMs and cancer cells), is also involved in this process. IDO is an enzyme that metabolizes tryptophan. The metabolites of this amino acid cause the differentiation of naïve CD4^+^ T cells into T_reg_ cells. In turn, T_reg_ cells produce IL-10, which enhances the functions of TADCs. In addition to their effects on T_reg_ cells, TADCs enhance tumor growth by producing HB-EGF and amphiregulin (activators of the EGFR receptors family).

**Table 1 ijms-21-07955-t001:** Prognosis for patients with increased CCL18 levels

Type of Cancer	Number of Patients Studied	Prognosis with Increased CCL18 Levels in the Tumor	Comments	Reference
Breast cancer (primary ductal carcinoma)	562	↓	CCL18^+^ TAM count	[69]
Breast cancer	1,017	↓	Serum level of CCL18 and expression in tumor	[108]
Breast cancer	207	↓	Serum level of CCL18	[173]
Breast phyllodes tumor	268	↓		[39]
Colorectal cancer	371	↑		[43]
Cutaneous T-cell lymphoma	38	↓	Serum level of CCL18	[148]
Gastric adenocarcinoma	90	↑		[44]
Laryngeal squamous cell carcinoma	146	↓	Serum level of CCL18	[174]
Lung cancer (adenocarcinoma)	170	↓	Serum level of CCL18	[176]
Lung cancer (non-small cell lung cancer)	80	↓	Serum level of CCL18	[175]
Lung cancer (non-small cell lung cancer)	243	↑	CCL18^+^ TAM count	[183]
Oral squamous cell carcinoma	102	↓		[182]
Ovarian cancer	59	↓		[55]
Ovarian cancer	187	↓		[177]
Osteosarcoma	102	↓	CCL18^+^ TAM count	[56]
Pancreatic ductal adenocarcinoma	62	↓	CCL18^+^ cells count	[59]
Pancreatic ductal adenocarcinoma	134	↓		[67]
Esophageal squamous cell carcinoma	25	↓		[86]

↑—better prognosis; ↓—poor prognosis.

**Table 2 ijms-21-07955-t002:** Prognosis for a patient with a given tumor with an increased expression of the CCL18 chemokine and its receptors, according to The Human Protein Atlas (https://www.proteinatlas.org) [184,185].

Type of Cancer	Prognosis with an Increased Expression of CCL18 in the Tumor	Prognosis with an Increased Expression of CCR6 in the Tumor	Prognosis with an Increased Expression of CCR8 in the Tumor	Prognosis with an Increased Expression of PITPNM3 in the Tumor	Prognosis with an Increased Expression of GPER1/GPR30 in the Tumor	Prognosis with an Increased Expression of GPER1/GPR30 in the Tumor (In Men)
Glioma	↓	↑ *p* = 0.076	↓	↓	--	--
Thyroid cancer	--	--	--	--	↓ *p* = 0.073	↓
Lung cancer	--	↑	↑	↑ *p* = 0.069	↑ *p* = 0.064	--
Colorectal cancer	↓ *p* = 0.057	↑	--	↑	↓	↓ *p* = 0.064
Head and neck cancer	↑	↑	↑	↑	↑ *p* = 0.062	↑ *p* = 0.093
Stomach cancer	--	↓	↑ *p* = 0.080	↑ *p* = 0.096	↓	↓
Liver cancer	--	--	--	↑	--	--
Pancreatic cancer	↓	↑	↑ *p* = 0.074	↓	↑	↑ *p* = 0.057
Renal cancer	↑	↓	↓	--	↑	↑
Urothelial cancer	↓	--	--	--	↓ *p* = 0.078	↓ *p* = 0.091
Prostate cancer	↓	↑	--	↓	--	--
Testis cancer	--	↓ *p* = 0.080	↓	--	--	--
Breast cancer	↑ *p* = 0.089	↑	--	↓	↑	N/A
Cervical cancer	↑	↑	↑	↑	--	N/A
Endometrial cancer	↑ *p* = 0.096	--	↑ *p* = 0.10	--	↑	N/A
Ovarian cancer	↑	--	↑	↓	--	N/A
Melanoma	--	--	↑	↓	--	--

↑—better prognosis; ↓—poor prognosis; --—no correlation

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
