# Peer review of "CCL18 in the Progression of Cancer"

_ijms, 2020, doi:10.3390/ijms21217955_

Round 1

Reviewer 1 Report

Authors provided the comprehensive data about the role of CCL18 in cancer. Considering CCL18 as a new target for anti-cancer treatment due to its pro-tumor properties, this review is relevent. 

However, Reviewer have noticed several part of the manuscript to be not adequate enough. The following comments have to be taken into account:

1) In the part 3 of the manuscript it is better to start with in vitro and in vivo studies, then consistently to give clinical data about the expression of CCL18 in different cancers. Please specifically indicate all data obtained on patient material.

2) Part 5 has to be "CCL18 as an inducer of EMT and migration of tumor cells"

3)It would imrove the understanding of mechanistical role of CCL18 in tumor progression if authors provide functions of CCL18 according to all its receptors consistently.

4) Page 7 - the part about the metabolism of tumor cells and NFkB is not sufficiantly clear here, as authors tell about the migration and EMT.

5)It is better to divide the expression of CCL18 in tumor cells and other cells of tumor microenvironment, such as TAMs.

6)Please give the discription about what are the main functions of mTOR? (page 7)

7) In part 8 authors should provide more info about the CCL18-expressing TAMs involved in tumor progression. AUthors can take for example the article "Expression of M2 macrophage markers YKL-39 and CCL18 in breast cancer is associated with the effect of neoadjuvant chemotherapy" (DOI: 10.1007/s00280-018-3594-8). There, CCL18-expressing subpopulation of TAMs is considered as a marker for chemotherapy response.

8) Mistakes: page 3 (conform to confirm), page 4 (CL18 to CCL18),page 8 (a lot of "which"), and others

Author Response

Authors provided the comprehensive data about the role of CCL18 in cancer. Considering CCL18 as a new target for anti-cancer treatment due to its pro-tumor properties, this review is relevent.

However, Reviewer have noticed several part of the manuscript to be not adequate enough. The following comments have to be taken into account:

1) In the part 3 of the manuscript it is better to start with in vitro and in vivo studies, then consistently to give clinical data about the expression of CCL18 in different cancers. Please specifically indicate all data obtained on patient material.

The article has been revised in line with the reviewer’s recommendations. At the beginning of the article there is information about the structure of the CCL18 gene and the CCL18 protein. Then we have included the results of the in vitro and in vivo experiments on the effects of CCL18. Then we have also included information about the experiments performed on cancer tissue specimens.

2) Part 5 has to be "CCL18 as an inducer of EMT and migration of tumor cells"

Corrected.

3)It would imrove the understanding of mechanistical role of CCL18 in tumor progression if authors provide functions of CCL18 according to all its receptors consistently.

Information on the consequences of the activation of receptors other than PITPNM3 by CCL18 has been added. Nevertheless, PubMed only offers a few publications on the participation of CCR3, CCR6, CCR8 and GPER1 / GPR30 in the operation of CCL18. We have emphasized this in our review.  

4) Page 7 - the part about the metabolism of tumor cells and NFkB is not sufficiantly clear here, as authors tell about the migration and EMT.

This excerpt has been moved to the TAM section as it covers the effects of CCL18 on these cells.

5)It is better to divide the expression of CCL18 in tumor cells and other cells of tumor microenvironment, such as TAMs.

Numerous in vivo studies show that TAMs are the major source of CCL18 in a tumour. CCL18 is considered as a marker for M2 macrophages. The tumour cell has a very low expression of this chemokine. We have emphasized these facts in the text.

  • Schutyser, E.; Struyf, S.; Proost, P.; Opdenakker, G.; Laureys, G.; Verhasselt, B.; Peperstraete, L.; Van de Putte, I.; Saccani, A.; Allavena, P.; Mantovani, A.; Van Damme, J. Identification of biologically active chemokine isoforms from ascitic fluid and elevated levels of CCL18/pulmonary and activation-regulated chemokine in ovarian carcinoma. J Biol Chem 2002, 277, 24584-24593.
  • Chang, C.Y.; Lee, Y.H.; Leu, S.J.; Wang, C.Y.; Wei, C.P.; Hung, K.S.; Pai, M.H.; Tsai, M.D.; Wu, C.H. CC-chemokine ligand 18/pulmonary activation-regulated chemokine expression in the CNS with special reference to traumatic brain injuries and neoplastic disorders. Neuroscience 2010, 165, 1233-1243.
  • Pettersen, J.S.; Fuentes-Duculan, J.; Suárez-Fariñas, M.; Pierson, K.C.; Pitts-Kiefer, A.; Fan, L.; Belkin, D.A.; Wang, C.Q.; Bhuvanendran, S.; Johnson-Huang, L.M.; Bluth, M.J.; Krueger, J.G.; Lowes, M.A.; Carucci, J.A. Tumor-associated macrophages in the cutaneous SCC microenvironment are heterogeneously activated. J Invest Dermatol 2011, 131, 1322-1330.
  • Yuan, R.; Chen, Y.; He, X.; Wu, X.; Ke, J.; Zou, Y.; Cai, Z.; Zeng, Y.; Wang, L.; Wang, J.; Fan, X.; Wu, X.; Lan, P. CCL18 as an independent favorable prognostic biomarker in patients with colorectal cancer. J Surg Res 2013, 183, 163-169.
  • Leung, S.Y.; Yuen, S.T.; Chu, K.M.; Mathy, J.A.; Li, R.; Chan, A.S.; Law, S.; Wong, J.; Chen, X.; So, S. Expression profiling identifies chemokine (C-C motif) ligand 18 as an independent prognostic indicator in gastric cancer. Gastroenterology 2004, 127, 457-469.

6)Please give the discription about what are the main functions of mTOR? (page 7)

A brief description of mTOR has been added.

7) In part 8 authors should provide more info about the CCL18-expressing TAMs involved in tumor progression. AUthors can take for example the article "Expression of M2 macrophage markers YKL-39 and CCL18 in breast cancer is associated with the effect of neoadjuvant chemotherapy" (DOI: 10.1007/s00280-018-3594-8). There, CCL18-expressing subpopulation of TAMs is considered as a marker for chemotherapy response.

CCL18+ TAM as a marker for chemotherapy response has been added to the section on anti-cancer therapy.

8) Mistakes: page 3 (conform to confirm), page 4 (CL18 to CCL18),page 8 (a lot of "which"), and others

The manuscript has been checked again and the errors found have been corrected.

Reviewer 2 Report

This manuscript by Korbecki et. al. is written well and comprehensively described the role of CCL18 in cancer progression. Authors should discuss about the therapeutic targeting of CCL18 in different cancers. Overall, very well written with good presentation.

Author Response

This manuscript by Korbecki et. al. is written well and comprehensively described the role of CCL18 in cancer progression. Authors should discuss about the therapeutic targeting of CCL18 in different cancers. Overall, very well written with good presentation.

A paragraph on CCL18 as a potential therapeutic target for non-breast cancer has been added. Nevertheless, there are no in vivo studies available on PubMed showing the therapeutic effect of blocking CCL18’s function.

Round 2

Reviewer 1 Report

All comments were adressed.